# Relational Reasoning via Set Transformers: Provable Efficiency and Applications to MARL

**Fengzhuo Zhang**[1]    **Boyi Liu**[2]    **Kaixin Wang**[1]    **Vincent Y. F. Tan**[1]    **Zhuoran Yang**[3]
**Zhaoran Wang**[2]

[1]National University of Singapore    [2] Northwestern University    [3]Yale University

`{fzzhang, kaixin.wang}@u.nus.edu, boyiliu2018@u.northwestern.edu,`
`vtan@nus.edu.sg, zhuoranyang.work@gmail.com, zhaoranwang@gmail.com`

## Abstract

The cooperative Multi-Agent Reinforcement Learning (MARL) with permutation invariant agents framework has achieved tremendous empirical successes in real-world applications. Unfortunately, the theoretical understanding of this MARL problem is lacking due to the curse of many agents and the limited exploration of the relational reasoning in existing works. In this paper, we verify that the transformer implements complex relational reasoning, and we propose and analyze model-free and model-based offline MARL algorithms with the transformer approximators. We prove that the suboptimality gaps of the model-free and model-based algorithms are independent of and logarithmic in the number of agents respectively, which mitigates the curse of many agents. These results are consequences of a novel generalization error bound of the transformer and a novel analysis of the Maximum Likelihood Estimate (MLE) of the system dynamics with the transformer. Our model-based algorithm is the first provably efficient MARL algorithm that explicitly exploits the permutation invariance of the agents. Our improved generalization bound may be of independent interest and is applicable to other regression problems related to the transformer beyond MARL.

## 1   Introduction

Cooperative MARL algorithms have achieved tremendous successes across a wide range of real-world applications including robotics [1, 2], games [3, 4], and finance [5]. In most of these works, the *permutation invariance* of the agents is embedded into the problem setup, and the successes of these works hinge on leveraging this property. However, the theoretical understanding of why the permutation invariant MARL has been so successful is lacking due to the following two reasons. First, the size of the state-action space grows exponentially with the number of agents; this is known as "the curse of many agents" [6, 7]. The exponentially large state-action space prohibits the learning of value functions and policies due to the curse of dimensionality. Second, although the mean-field approximation is widely adopted to mitigate the curse of many agents [6, 8], this approximation fails to capture the complex interplay between the agents. In the mean-field approximation, the influence of all the other agents on a fixed agent is captured only through the empirical distribution of the local states and/or local actions [6, 8]. This induces a restricted class of function approximators, which nullifies the possibly complicated relational structure of the agents, and thus fails to incorporate the complex interaction between agents. Therefore, designing provably efficient MARL algorithms that incorporate the efficient relational reasoning and break the curse of many agents remains an interesting and meaningful question.

In this paper, we regard transformer networks as the representation learning module to incorporate relational reasoning among the agents. In particular, we focus on the offline MARL problem with

36th Conference on Neural Information Processing Systems (NeurIPS 2022).

the transformer approximators in the cooperative setting. In this setting, all the agents learn policies *cooperatively* to maximize a common reward function. More specifically, in the offline setting, the learner only has access to a pre-collected dataset and cannot interact adaptively with the environment. Moreover, we assume that the underlying Markov Decision Process (MDP) is *homogeneous*, which means that the reward and the transition kernel are permutation invariant functions of the state-action pairs of the agents. Our goal is to learn an *optimal* policy that is also permutation invariant.

To design provably efficient offline MARL algorithms, we need to overcome three key challenges. (i) To estimate the action-value function and the system dynamics, the approximator function needs to implement efficient relational reasoning among the agents. However, the theoretically-grounded function structure that incorporates the complex relational reasoning needs to be carefully designed. (ii) To mitigate the curse of many agents, the generalization bound of the transformer should be independent of the number of agents. Existing results in [9] thus require rethinking and improvements. (iii) In offline Reinforcement Learning (RL), the mismatch between the sampling and visitation distributions induced by the optimal policy (i.e., "distribution shift") greatly restricts the application of the offline RL algorithm. Existing works adopt the *"pessimism"* principle to mitigate such a challenge. However, this requires the quantification of the uncertainty in the value function estimation and the estimation of the dynamics in the model-free and model-based methods respectively. The quantification of the estimation error with the transformer function class is a key open question.

We organize our work by addressing the abovementioned three challenges.

First, we theoretically identify the function class that can implement complex relational reasoning. We demonstrate the relational reasoning ability of the attention mechanism by showing that approximating the self-attention structure with the permutation invariant fully-connected neural networks (i.e., deep sets [10]) requires an *exponentially large* number of hidden nodes in the input dimension of each channel (Theorem 1). This result necessitates the self-attention structure in the set transformer.

Second, we design offline model-free and model-based RL algorithms with the transformer approximators. In the former, the transformer is adopted to estimate the action-value function of the policy. The *pessimism* is encoded in that we learn the policy according to the *minimal* estimate of the action-value function in the set of functions with bounded empirical Bellman error. In the model-based algorithm, we estimate the system dynamics with the transformer structure. The policy is learned pessimistically according to the estimate of the system dynamics in the confidence region that induces the conservative value function.

Finally, we analyze the suboptimality gaps of our proposed algorithms, which indicate that the proposed algorithms mitigate the curse of many agents. For the model-free algorithm, the suboptimality gap in Theorem 3 is independent of the number of agents, which is a consequence of the fact that the generalization bound of the transformer (Theorem 2) is independent of the number of channels. For the model-based algorithm, the bound on the suboptimality gap in Theorem 4 is logarithmic in the number of agents; this follows from the analysis of the MLE of the system dynamics in Proposition 3. We emphasize that our model-based algorithm is the first provably efficient MARL algorithm that exploits the permutation equivariance when estimating the dynamics.

**Technical Novelties.** In Theorem 2, we leverage a PAC-Bayesian framework to derive a generalization error bound of the transformer. Compared to [9, Theorem 4.6], the result is a significant improvement in the dependence on the number of channels $N$ and the depth of neural network $L$. This result may be of independent interest for enhancing our theoretical understanding of the attention mechanism and is applicable to other regression problems related to the transformer. In Proposition 3, we derive the first estimation uncertainty quantification of the system dynamics with the transformer approximators, which can be also be used to analyze other RL algorithms with such approximators.

**More Related Work.** In this paper, we consider the offline RL problem, and the insufficient coverage lies at the core of this problem. With the global coverage assumption, a number of works have been proposed from both the model-free [11–15] and model-based [11, 16] perspectives. To weaken the global coverage assumption, we leverage the "pessimism" principle in the algorithms: the model-free algorithms impose additional penalty terms on the estimate of the value function [17, 18] or regard the function that attains the minimum in the confidence region as the estimate of the value function [19]; the model-based algorithms estimate the system dynamics by incorporating additional penalty terms [20] or minimizing in the region around MLE [21]. For the MARL setting, the offline MARL with the mean-field approximation has been studied in [8, 22].

The analysis of the MARL algorithm with the transformer approximators requires the generalization bound of the transformer. The transformer is an element of the group equi/invariant functions, whose benefit in terms of its generalization capabilities has attracted extensive recent attention. Generalization bounds have been successively improved by analyzing the cardinality of the "effective" input field and Lipschitz constants of functions [23, 24]. However, these methods result in loose generalization bounds when applied to deep neural networks [25]. Zhu, An, and Huang [26] empirically demonstrated the benefits of the invariance in the model by refining the covering number of the function class, but a unified theoretical understanding is still lacking. The covering number of the norm-bounded transformer was shown by [9] to be at most logarithmic in the number of channels. We show that this can be further improved using a PAC-Bayesian framework. In addition, we refer to the related concurrent work [27] for a Rademacher complexity-based generalization bound of the transformer that is independent of the length of the sequence for the tasks such as computer vision.

## 2 Preliminaries

**Notation.** Let $[n] = \{1, \ldots, n\}$. The $i^{\text{th}}$ entry of the vector $x$ is denoted as $x_i$ or $[x]_i$. The $i^{\text{th}}$ row and the $i^{\text{th}}$ column of matrix $X$ are denoted as $X_{i,:}$ and $X_{:,i}$ respectively. The $\ell_p$-*norm* of the vector $x$ is $\|x\|_p$. The $\ell_{p,q}$-*norm* of the matrix $X \in \mathbb{R}^{m \times n}$ is defined as $\|X\|_{p,q} = (\sum_{i=1}^{n} \|X_{:,i}\|_p^q)^{1/q}$, and the *Frobenius norm* of $X$ is defined as $\|X\|_{\text{F}} = \|X\|_{2,2}$. The *total variation distance* between two distributions $P$ and $Q$ on $\mathcal{A}$ is defined as $\text{TV}(P,Q) = \sup_{A \subseteq \mathcal{A}} |P(A) - Q(A)|$. For a set $\mathcal{X}$, we use $\Delta(\mathcal{X})$ to denote the set of distributions on $\mathcal{X}$. For two conditional distributions $P, Q : \mathcal{X} \rightarrow \Delta(\mathcal{Y})$, the $d_\infty$ distance between them is defined as $d_\infty(P,Q) = 2 \sup_{x \in \mathcal{X}} \text{TV}(P(\cdot \mid x), Q(\cdot \mid x))$. Given a metric space $(\mathcal{X}, \|\cdot\|)$, for a set $\mathcal{A} \subseteq \mathcal{X}$, an $\varepsilon$-*cover* of $\mathcal{A}$ is a finite set $\mathcal{C} \subseteq \mathcal{X}$ such that for any $a \in \mathcal{A}$, there exists $c \in \mathcal{C}$ and $\|c - a\| \leq \varepsilon$. The $\varepsilon$-*covering number* of $\mathcal{A}$ is the cardinality of the smallest $\varepsilon$-cover, which is denoted as $\mathcal{N}(\mathcal{A}, \varepsilon, \|\cdot\|)$.

**Attention Mechanism and Transformers.** The *attention mechanism* is a technique that mimics cognitive attention to process multi-channel inputs [28]. Compared with the Convolutional Neural Network (CNN), the transformer has been empirically shown to possess outstanding robustness against occlusions and preserve the global context due to its special relational structure [29]. Assume we have $N$ query vectors that are in $\mathbb{R}^{d_Q}$. These vectors are stacked to form the matrix $Q \in \mathbb{R}^{N \times d_Q}$. With $N_V$ *key vectors* in the matrix $K \in \mathbb{R}^{N_V \times d_Q}$ and $N_V$ *value vectors* in the matrix $V \in \mathbb{R}^{N_V \times d_V}$, the attention mechanism maps the queries $Q$ using the function $\text{Att}(Q, K, V) = \text{SM}(QK^\top)V$, where $\text{SM}(\cdot)$ is the row-wise softmax operator that normalizes each row using the exponential function, i.e., for $x \in \mathbb{R}^d$, $[\text{SM}(x)]_i = \exp(x_i)/\sum_{j=1}^{d} \exp(x_j)$ for $i \in [d]$. The product $QK^\top$ measures the similarity between the queries and the keys, which is then passed through the activation function $\text{SM}(\cdot)$. Thus, $\text{SM}(QK^\top)V$ essentially outputs a weighted sum of $V$ where a value vector has greater weight if the corresponding query and key are more similar. The *self-attention mechanism* is defined as the attention that takes $Q = XW_Q$, $K = XW_K$ and $V = XW_V$ as inputs, where $X \in \mathbb{R}^{N \times d}$ is the input of self-attention, and $W_Q, W_K \in \mathbb{R}^{d \times d_Q}$ and $W_V \in \mathbb{R}^{d \times d_V}$ are the parameters. Intuitively, self-attention weighs the inputs with the correlations among $N$ different channels. This mechanism demonstrates a special pattern of *relational reasoning* among the channels of $X$.

In addition, the self-attention mechanism is *permutation invariant* in the channels in $X$. This implies that for any row-wise permutation function $\psi(\cdot)$, which swaps the rows of the input matrix according to a given permutation of $[N]$, we have $\text{Att}(\psi(X)W_Q, \psi(X)W_K, \psi(X)W_V) = \psi(\text{Att}(XW_Q, XW_K, XW_V))$. The permutation equivariance of the self-attention renders it suitable for inference tasks where the output is equivariant with respect to the ordering of inputs. For example, in image segmentation, the result should be invariant to the permutation of the objects in the input image [30]. The resultant transformer structure combines the self-attention with multi-layer perceptrons and composes them to form deep neural networks. It remains permutation equi/invariant with respect to the order of the channels and has achieved excellent performance in many tasks [31–33].

**Offline Cooperative MARL.** In this paper, we consider the *cooperative* MARL problem, where all agents aim to maximize a *common* reward function. The corresponding MDP is characterized by the tuple $(\bar{S}_0, \bar{\mathcal{S}}, \bar{\mathcal{A}}, P^*, r, \gamma)$ and the number of agents is $N$. The state space $\bar{\mathcal{S}} = \mathcal{S}^N$ is the Cartesian product of the state spaces of each agent $\mathcal{S}$, and $\bar{S} = [s_1, \ldots, s_N]^\top$ is the state, where $s_i \in \mathbb{R}^{d_\mathcal{S}}$ is the state of the $i^{\text{th}}$ agent. The initial state is $\bar{S}_0$. The action space $\bar{\mathcal{A}} = \mathcal{A}^N$ is the Cartesian product of the action spaces $\mathcal{A}$ of each agent, and $\bar{A} = [a_1, \ldots, a_N]^\top$ is the action, where

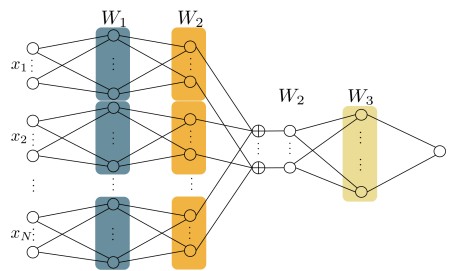



(a) $\rho_{\mathrm{ReLU}}(\sum_{i=1}^N \phi_{\mathrm{ReLU}}(x_i))$ with $\rho_{\mathrm{ReLU}}$ and $\psi_{\mathrm{ReLU}}$ as single-hidden layer neural networks.

(b) Self-attention mechanism $\mathbb{I}_N^\top \mathrm{Att}(\mathrm{X,X,X})\mathrm{w}$.

Figure 1: The blocks with the same color share the same parameters. The left figure shows that $\rho_{\mathrm{ReLU}}(\sum_{i=1}^N \phi_{\mathrm{ReLU}}(x_i))$ first sums the outputs of $\phi_{\mathrm{ReLU}}(x_i)$, and it implements the relational reasoning only through the single-hidden layer network $\rho_{\mathrm{ReLU}}$. In contrast, the self-attention block in the right figure captures the relationship among channels and then sums the outputs of each channel.

$a_i \in \mathbb{R}^{d_{\mathcal{A}}}$ is the action of the $i^{\mathrm{th}}$ agent. The transition kernel is $P^* : \mathcal{S}^N \times \mathcal{A}^N \to \Delta(\mathcal{S}^N)$, and $\gamma \in (0,1)$ is the *discount factor*. Without loss of generality, we assume that the reward function $r$ is deterministic and bounded, i.e., $r : \mathcal{S}^N \times \mathcal{A}^N \to [-R_{\max}, R_{\max}]$. We define the the *state-value function* $V_P^\pi : \mathcal{S}^N \to [-V_{\max}, V_{\max}]$, where $V_{\max} = R_{\max}/(1-\gamma)$, and the *action-value function* $Q_P^\pi : \mathcal{S}^N \times \mathcal{A}^N \to [-V_{\max}, V_{\max}]$ of a policy $\pi$ and a transition kernel $P$ as

$$V_P^\pi(\bar{S}) = \mathbb{E}^\pi \left[ \sum_{t=0}^\infty \gamma^t r(\bar{S}_t, \bar{A}_t) \,\middle|\, \bar{S}_0 = \bar{S} \right] \text{ and } Q_P^\pi(\bar{S}, \bar{A}) = \mathbb{E}^\pi \left[ \sum_{t=0}^\infty \gamma^t r(\bar{S}_t, \bar{A}_t) \,\middle|\, \bar{S}_0 = \bar{S}, \bar{A}_0 = \bar{A} \right],$$

respectively. Here, the expectation is taken with respect to the Markov process induced by the policy $\bar{A}_t \sim \pi(\cdot \,|\, \bar{S}_t)$ and the transition kernel $P$. The action-value function $Q_{P^*}^\pi$ is the unique fixed point of the operator $(\mathcal{T}^\pi f)(\bar{S}, \bar{A}) = r(\bar{S}, \bar{A}) + \gamma \mathbb{E}_{\bar{S}' \sim P^*(\cdot \,|\, \bar{S}, \bar{A})}[f(\bar{S}', \pi) \,|\, \bar{S}, \bar{A}]$, where the term in the expectation is defined as $f(\bar{S}, \pi) = \mathbb{E}_{\bar{A} \sim \pi(\cdot \,|\, \bar{S})}[f(\bar{S}, \bar{A})]$. We further define the *visitation measure* of the state and action pair induced the policy $\pi$ and transition kernel $P$ as $d_P^\pi(\bar{S}, \bar{A}) = (1-\gamma) \sum_{t=0}^\infty \gamma^t d_{P,t}^\pi$, where $d_{P,t}^\pi$ is the distribution of the state and the action at step $t$.

In offline RL, the learner only has access to a pre-collected dataset and cannot interact with the environment. The dataset $\mathcal{D} = \{(\bar{S}_i, \bar{A}_i, r_i, \bar{S}'_i)\}_{i=1}^n$ is collected in an i.i.d. manner, i.e., $(\bar{S}_i, \bar{A}_i)$ is independently sampled from $\nu \in \Delta(\bar{\mathcal{S}} \times \bar{\mathcal{A}})$, and $\bar{S}'_i \sim P^*(\cdot \,|\, \bar{S}_i, \bar{A}_i)$. This i.i.d. assumption is made to simplify our theoretical results; see Appendix N.2 for extensions to the non i.i.d. case. Given a policy class $\Pi$, our goal is to find an optimal policy that maximizes the state-value function $\pi^* = \arg\max_{\pi \in \Pi} V_{P^*}^\pi(\bar{S}_0)$. For any $\pi \in \Pi$, the *suboptimality gap* of $\pi$ is defined as $V_{P^*}^{\pi^*}(\bar{S}_0) - V_{P^*}^\pi(\bar{S}_0)$.

## 3 Provable Efficiency of Transformer on Relational Reasoning

In this section, we provide the theoretical understanding of the outstanding relational reasoning ability of transformer. These theoretical results serves as a firm base for adopting set transformer to estimate the value function and system dynamics in RL algorithms in the following sections.

### 3.1 Relational Reasoning Superiority of Transformer Over MLP

The transformer neural network combines the self-attention mechanism and the fully-connected neural network, which includes the MultiLayer Perceptrons (MLP) function class as a subset. On the inverse direction, we show that permutation invariant MLP can not approximate transformer unless its width is exponential in the input dimension due to the poor relational reasoning ability of MLP.

Zaheer et al. [10, Theorem 2] showed that all permutation invariant functions take the form $\rho(\sum_{i=1}^N \phi(x_i))$ with $X = [x_1, \ldots, x_N]^\top \in \mathbb{R}^{N \times d}$ as the input. Since the single-hidden layer ReLU neural network is an universal approximator for continuous functions [34], we set $\phi : \mathbb{R}^{N \times d} \to \mathbb{R}^{W_2}$ and $\rho : \mathbb{R}^{W_2} \to \mathbb{R}$ to be single-hidden layer neural networks with ReLU activation functions as shown in Figure 1(a), where $W_2$ is the dimension of the intermediate outputs. The widths of the hidden layers in $\phi_{\mathrm{ReLU}}$ and $\rho_{\mathrm{ReLU}}$ are $W_1$ and $W_3$ respectively. For the formal definition of $\phi_{\mathrm{ReLU}}$ and $\rho_{\mathrm{ReLU}}$,

please refer to Appendix A. Then the function class with $\rho_{\text{ReLU}}$ and $\phi_{\text{ReLU}}$ as width-constrained ReLU networks is defined as

$$\mathcal{N}(W) = \left\{ f : \mathbb{R}^{N \times d} \to \mathbb{R} \,\middle|\, f(X) = \rho_{\text{ReLU}}\left( \sum_{i=1}^{N} \phi_{\text{ReLU}}(x_i) \right) \text{ with } \max_{i \in [3]} W_i \le W \right\}.$$

We would like to use functions in $\mathcal{N}(W)$ to approximate the self-attention function class

$$\mathcal{F} = \left\{ f : \mathbb{R}^{N \times d} \to \mathbb{R} \,\middle|\, f(X) = \mathbb{I}_N^\top \text{Att}(X, X, X) w \text{ for some } w \in [0,1]^d \right\}.$$

Figure 1(a) shows that $\rho_{\text{ReLU}}(\sum_{i=1}^{N} \phi_{\text{ReLU}}(x_i))$ first processes each channel with $\phi_{\text{ReLU}}$, and the relationship between channels is only reasoned with $\rho_{\text{ReLU}}$. The captured relationship in $\rho_{\text{ReLU}}(\sum_{i=1}^{N} \phi_{\text{ReLU}}(x_i))$ cannot be too complex due to the simple structure of $\rho_{\text{ReLU}}$. In contrast, the self-attention structure shown in Figure 1(b) first captures the relationship between channels with the self-attention structure and then weighs the results to derive the final output. Consequently, it is difficult to approximate the self-attention structure with $\rho_{\text{ReLU}}(\sum_{i=1}^{N} \phi_{\text{ReLU}}(x_i))$ due to its poor relational reasoning ability. This observation is formally quantified in the following theorem.

**Theorem 1.** *Let $W^*(\xi, d, \mathcal{F})$ be the smallest width of the neural network such that*

$$\forall f \in \mathcal{F}, \ \exists g \in \mathcal{N}(W) \quad s.t. \quad \sup_{X \in [0,1]^{N \times d}} \left| f(X) - g(X) \right| \le \xi.$$

*With sufficient number of channels $N$, it holds that $W^*(\xi, d, \mathcal{F}) = \Omega(\exp(cd)\xi^{-1/4})$ for some $c > 0$.*

Theorem 1 shows that the fully-connected neural network cannot approximate the relational reasoning process in the self-attention mechanism unless the width is exponential in the input dimension. This exponential lower bound of the width of the fully-connected neural network implies that the relational reasoning process embedded within the self-attention structure is complicated, and it further motivates us to explicitly incorporate the self-attention structure in the neural networks in order to reason the complex relationship among the channels.

## 3.2 Channel Number-independent Generalization Error Bound

In this section, we derive the generalization error bound of transformer. We take $X \in \mathbb{R}^{N \times d}$ as the input of the neural network. In the $i^{\text{th}}$ layer, as shown in Figure 3.2, we combine the self-attention mechanism $\text{Att}(XW_{QK}^{(i)}, X, XW_V^{(i)})$ with the row-wise FeedForward (rFF) single-hidden layer neural network $\text{rFF}(X, a^{(i)}, b^{(i)})$ with width $m$. We combine $W_Q^{(i)}$ and $W_K^{(i)}$ to $W_{QK}^{(i)}$ for ease of calculation, and $b^{(i)}$ and $a^{(i)}$ are the parameters of the first and second layer of rFF. The output of each layer is normalized by the row-wise normalization function $\Pi_{\text{norm}}(\cdot)$, which projects each row of the input into the unit $\ell_p$-ball (for some $p \ge 1$). For the last layer, we derive

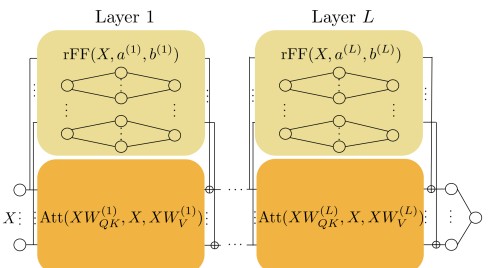

Figure 2: Structure of the transformer function class, where the row-wise feedforward function is specified as fully-connected networks.

the scalar estimate of the action-value function by averaging the outputs of all the channels, and the "clipping" function $\Pi_V(x)$ is applied to normalize the output to $[-V, V]$. We note that such structures are also known as *set transformers* in [33]. For the formal definition of the transformer, please refer to Appendix B.

We consider a transformer with bounded parameters. For a pair of conjugate numbers $p, q \in \mathbb{R}$, i.e., $1/p + 1/q = 1$ and $p, q \ge 1$, the transformer function class with bounded parameters is defined as

$$\mathcal{F}_{\text{tf}}(B) = \left\{ g_{\text{tf}}(X; W_{QK}^{1:L}, W_V^{1:L}, a^{1:L}, b^{1:L}, w) \,\middle|\, \left| a_{kj}^{(i)} \right| < B_a, \left\| b_{kj}^{(i)} \right\|_q < B_b, \right.$$

$$\left. \left\| W_{QK}^{(i)\top} \right\|_{p,q} < B_{QK}, \left\| W_V^{(i)\top} \right\|_{p,q} < B_V, \|w\|_q < B_w \text{ for } i \in [L], j \in [m], k \in [d] \right\},$$

where $B = [B_a, B_b, B_{QK}, B_V, B_w]$ are the parameters of the function class, and $W_{QK}^{1:L}, W_V^{1:L}, a^{1:L}$ and $b^{1:L}$ are the stacked parameters in each layer. We only consider the non-trivial case where

$B_a, B_b, B_{QK}, B_V, B_w$ are larger than one, otherwise the norms of the outputs decrease exponentially with growing depth. For ease of notation, we denote $\mathcal{F}_{\mathrm{tf}}(B)$ as $\mathcal{F}_{\mathrm{tf}}$ when the parameters are clear.

Consider the regression problem where we aim to predict the value of the response variable $y \in \mathbb{R}$ from the observation matrix $X \in \mathbb{R}^{N \times d}$, where $(X, y) \sim \nu$, and $|y| \leq V$. We derive our estimate $f : \mathbb{R}^{N \times d} \to \mathbb{R}$ from i.i.d. observations $\mathcal{D}_{\mathrm{reg}} = \{(X_i, y_i)\}_{i=1}^{n}$ generated from $\nu$. The *risk* of using $f \in \mathcal{F}_{\mathrm{tf}}(B)$ as a regressor on sample $(X, y)$ is defined as $(f(X) - y)^2$. Then the *excess risk* of functions in the transformer function class $\mathcal{F}_{\mathrm{tf}}$ can be bounded as in the following proposition.

**Proposition 1.** *Let $\bar{B} = B_V B_{QK} B_a B_b B_w$. For all $f \in \mathcal{F}_{\mathrm{tf}}$, with probability at least $1 - \delta$, we have*

$$\left| \mathbb{E}_\nu \Big[ \big( f(X) - y \big)^2 \Big] - \frac{1}{n} \sum_{i=1}^{n} \big( f(X_i) - y_i \big)^2 \right|$$

$$\leq \frac{1}{2} \mathbb{E}_\nu \Big[ \big( f(X) - y \big)^2 \Big] + O\left( \frac{V^2}{n} \left[ mL^2 d^2 \log \frac{mdL\bar{B}n}{V} + \log \frac{1}{\delta} \right] \right).$$

Proposition 1 is a corollary of Theorem 2. We state it here since the generalization error bound of transformer may be interesting for other regression problems. We compare our generalization error bound in Proposition 1 with [9, Theorem 4.6]. For the dependence on the number of agents $N$, the result in [9, Theorem 4.6] shows that the logarithm of the covering number of the transformer function class is logarithmic in $N$. Combined with the use of the Dudley integral [35], [9, Theorem 4.6] implies that the generalization error bound is logarithmic in $N$. In contrast, our result is independent of $N$. This superiority is attributed to our use of the PAC-Bayesian framework, in which we measure the distance between functions using the KL divergence of the distributions on the function parameter space. For the transformer structure, the size of the parameter space is independent of the number of agents $N$, which helps us to remove the dependence on $N$.

Concerning the dependence on the depth $L$ of the neural network, [9, Theorem 4.6] shows that the logarithm of the covering number of the transformer function class scales exponentially in $L$. In contrast, Proposition 1 shows that the generalization bound is *polynomial* in $L$. We note that Proposition 1 does not contradict the exponential dependence shown in [36, 37], since we implement the layer normalization to restrict the range of the output. As a byproduct, Proposition 1 shows that the invariant of the layer normalization adopted in our paper can greatly reduce the dependence of the generalization error on the depth of the neural network $L$. We note that our results can be generalized to the multi-head attention structure, and the extensions are provided in Appendix N.

## 4 Offline Multi-Agent Reinforcement Learning with Set Transformers

In this section, we apply the results in Section 3 to MARL. We implement efficient relational reasoning via the set transformer to obtain improved suboptimality bounds of the MARL problem. In particular, we consider the *homogeneous* MDP, where the transition kernel and the reward function are invariant to permutations of the agents, i.e., for any row-wise permutation function $\psi(\cdot)$, we have

$$P^*(\bar{S}' \,|\, \bar{S}, \bar{A}) = P^*\big(\psi(\bar{S}') \,\big|\, \psi(\bar{S}), \psi(\bar{A})\big) \quad \text{and} \quad r(\bar{S}, \bar{A}) = r\big(\psi(\bar{S}), \psi(\bar{A})\big)$$

for all $\bar{S}, \bar{S}' \in \mathcal{S}^N$ and $\bar{A} \in \mathcal{A}^N$. A key property of the homogeneous MDP is that there exists a permutation invariant optimal policy, and the corresponding state-value function and the action-value function are also permutation invariant [22].

**Proposition 2.** *For the cooperative homogeneous MDP, there exists an optimal policy that is permutation invariant. Also, for any permutation invariant policy $\pi$, the corresponding value function $V_{P^*}^\pi$ and action-value function $Q_{P^*}^\pi$ are permutation invariant.*

Thus, we restrict our attention to the class of permutation invariant policies $\Pi$, where $\pi(\bar{A} \,|\, \bar{S}) = \pi(\psi(\bar{A}) \,|\, \psi(\bar{S}))$ for all $\bar{A} \in \bar{\mathcal{A}}$, $\bar{S} \in \bar{\mathcal{S}}$, $\pi \in \Pi$ and all permutations $\psi$. For example, if $\pi(\bar{A} \,|\, \bar{S}) = \prod_{i=1}^{N} \mu(a_i \,|\, s_i)$ for some $\mu$, then $\pi$ is permutation invariant. An optimal policy is any $\pi^* \in \mathrm{argmax}_{\pi \in \Pi} V_{P^*}^\pi(\bar{S}_0)$.

### 4.1 Pessimistic Model-Free Offline Reinforcement Learning

In this subsection, we present a model-free algorithm, in which we adopt the transformer to estimate the action-value function. We also learn a policy based on such an estimate.

#### 4.1.1 Algorithm

We modify the single-agent offline RL algorithm in [19] to be applicable to the multi-agent case with the transformer approximators, but the analysis is rather different from that in [19]. Given the dataset $\mathcal{D} = \{(\bar{S}_i, \bar{A}_i, r_i, \bar{S}'_i)\}_{i=1}^n$, we define the mismatch between two functions $f$ and $\tilde{f}$ on $\mathcal{D}$ for a fixed policy $\pi$ as $\mathcal{L}(f, \tilde{f}, \pi; \mathcal{D}) = \frac{1}{n} \sum_{(\bar{S}, \bar{A}, \bar{r}, \bar{S}') \in \mathcal{D}} (f(\bar{S}, \bar{A}) - \bar{r} - \gamma \tilde{f}(\bar{S}', \pi))^2$. We adopt the transformer function class $\mathcal{F}_{\mathrm{tf}}(B)$ in Section 3.2 to estimate the action-value function and regard $X = [\bar{S}, \bar{A}] \in \mathbb{R}^{N \times d}$ as the input of the neural network. The dimension $d = d_{\mathcal{S}} + d_{\mathcal{A}}$ and each agent corresponds to a channel in $X$. The *Bellman error* of a function $f$ with respect to the policy $\pi$ is defined as $\mathcal{E}(f, \pi; \mathcal{D}) = \mathcal{L}(f, f, \pi; \mathcal{D}) - \inf_{\tilde{f} \in \mathcal{F}_{\mathrm{tf}}} \mathcal{L}(\tilde{f}, f, \pi; \mathcal{D})$.

For a fixed policy $\pi$, we construct the confidence region of the action-value function of $\pi$ by selecting the functions in $\mathcal{F}_{\mathrm{tf}}$ with the $\varepsilon$-controlled Bellman error. We regard the function attaining the minimum in the confidence region as the estimate of the action-value function of the policy; this reflects the terminology "pessimism". Then the optimal policy is learned by maximizing the action-value function estimate. The algorithm can be written formally as

$$\hat{\pi} = \arg\max_{\pi \in \Pi} \min_{f \in \mathcal{F}(\pi, \varepsilon)} f(\bar{S}_0, \pi), \quad \text{where} \quad \mathcal{F}(\pi, \varepsilon) = \left\{ f \in \mathcal{F}_{\mathrm{tf}}(B) \,\middle|\, \mathcal{E}(f, \pi; \mathcal{D}) \leq \varepsilon \right\}. \quad (1)$$

The motivation for the pessimism originates from the *distribution shift*, where the induced distribution of the learned policy is different from the sampling distribution $\nu$. Such an issue is severe when there is no guarantee that the sampling distribution $\nu$ supports the visitation distribution $d_{P*}^{\pi^*}$ induced by the optimal policy $\pi^*$. In fact, the algorithm in Eqn. (1) does not require the *global* coverage of the sampling distribution $\nu$, where the global coverage means that $d_{P*}^{\pi^*}(\bar{S}, \bar{A})/\nu(\bar{S}, \bar{A})$ is upper bounded by some constant for all $(\bar{S}, \bar{A}) \in \bar{\mathcal{S}} \times \bar{\mathcal{A}}$ and all $\pi \in \Pi$. Instead, it only requires *partial* coverage, and the mismatch between the distribution induced by the optimal policy $d_{P*}^{\pi^*}$ and the sampling distribution $\nu$ is captured by

$$C_{\mathcal{F}_{\mathrm{tf}}} = \max_{f \in \mathcal{F}_{\mathrm{tf}}} \mathbb{E}_{d_{P*}^{\pi^*}} \left[ \left( f(\bar{S}, \bar{A}) - \mathcal{T}^{\pi^*} f(\bar{S}, \bar{A}) \right)^2 \right] \Big/ \mathbb{E}_{\nu} \left[ \left( f(\bar{S}, \bar{A}) - \mathcal{T}^{\pi^*} f(\bar{S}, \bar{A}) \right)^2 \right]. \quad (2)$$

We note that $C_{\mathcal{F}_{\mathrm{tf}}} \leq \max_{(\bar{S}, \bar{A}) \in \bar{\mathcal{S}} \times \bar{\mathcal{A}}} d_{P*}^{\pi^*}(\bar{S}, \bar{A})/\nu(\bar{S}, \bar{A})$, so the suboptimality bound involving $C_{\mathcal{F}_{\mathrm{tf}}}$ in Theorem 3 is tighter than the bound requiring global convergence [38]. Similar coefficients also appear in many existing works such as [19] and [39].

#### 4.1.2 Bound on the Suboptimality Gap

Before stating the suboptimality bound, We require two assumptions on $\mathcal{F}_{\mathrm{tf}}$ and the sampling distribution $\nu$. We first state the standard regularity assumption of the transformer function class.

**Assumption 1.** *For any $\pi \in \Pi$, we have $\inf_{f \in \mathcal{F}_{\mathrm{tf}}} \sup_{\mu \in d_\Pi} \mathbb{E}_\mu[(f(\bar{S}, \bar{A}) - \mathcal{T}^\pi f(\bar{S}, \bar{A}))^2] \leq \varepsilon_{\mathcal{F}}$ and $\sup_{f \in \mathcal{F}_{\mathrm{tf}}} \inf_{\tilde{f} \in \mathcal{F}_{\mathrm{tf}}} \mathbb{E}_\nu[(\tilde{f}(\bar{S}, \bar{A}) - \mathcal{T}^\pi f(\bar{S}, \bar{A}))^2] \leq \varepsilon_{\mathcal{F}, \mathcal{F}}$, where $d_\Pi = \{\mu \mid \exists \pi \in \Pi \ s.t. \ \mu = d_{P*}^\pi\}$ is the set of distributions of the state and the action pair induced by any policy $\pi \in \Pi$.*

This assumption, including the *realizability* and the *completeness*, states that for any policy $\pi \in \Pi$ there is a function in the transformer function class $\mathcal{F}_{\mathrm{tf}}$ such that the Bellman error is controlled by $\varepsilon_{\mathcal{F}}$, and the transformer function class is approximately closed under the Bellman operator $\mathcal{T}^\pi$ for any $\pi \in \Pi$. In addition, we require that the mismatch between the sampling distribution and the visitation distribution of the optimal policy is bounded.

**Assumption 2.** *For the sampling distribution $\nu$, the coefficient $C_{\mathcal{F}_{\mathrm{tf}}}$ defined in Eqn. (2) is finite.*

We note that similar assumptions also appear in many existing works [19, 39].

In the analysis of the algorithm in Eqn. (1), we first derive a generalization error bound of the estimate of the Bellman error using the PAC-Bayesian framework [40, 41].

**Theorem 2.** *Let $\bar{B} = B_V B_{QK} B_a B_b B_w$. For all $f, \tilde{f} \in \mathcal{F}_{\mathrm{tf}}(B)$ and all policies $\pi \in \Pi$, with probability at least $1 - \delta$, we have*

$$\left| \mathbb{E}_\nu \left[ \left( f(\bar{S}, \bar{A}) - \mathcal{T}^\pi \tilde{f}(\bar{S}, \bar{A}) \right)^2 \right] - \mathcal{L}(f, \tilde{f}, \pi; \mathcal{D}) + \mathcal{L}(\mathcal{T}^\pi \tilde{f}, \tilde{f}, \pi; \mathcal{D}) \right|$$

$$\leq \frac{1}{2} \mathbb{E}_\nu \left[ \left( f(\bar{S}, \bar{A}) - \mathcal{T}^\pi \tilde{f}(\bar{S}, \bar{A}) \right)^2 \right] + O\left( \frac{V_{\max}^2}{n} \left[ mL^2 d^2 \log \frac{mdL\bar{B}n}{V_{\max}} + \log \frac{\mathcal{N}(\Pi, 1/n, d_\infty)}{\delta} \right] \right).$$

For ease of notation, we define $e(\mathcal{F}_{\mathrm{tf}}, \Pi, \delta, n)$ to be $n$ times the second term of the generalization error bound. We note that the generalization error bound in Theorem 2 is independent of the number of agents, which will help us to remove the dependence on the number of agents in the suboptimality of the learned policy. The suboptimality gap of the learned policy $\hat{\pi}$ can be upper bounded as the following.

**Theorem 3.** *If Assumptions 1 and 2 hold, and we take $\varepsilon = 3\varepsilon_{\mathcal{F}}/2 + 2e(\mathcal{F}_{\mathrm{tf}}, \Pi, \delta, n)/n$, then with probability at least $1 - \delta$, the suboptimality gap of the policy derived in the algorithm shown in Eqn. (1) is upper bounded as*

$$V_{P^*}^{\pi^*}(\bar{S}_0) - V_{P^*}^{\hat{\pi}}(\bar{S}_0) \leq O\left( \frac{\sqrt{C_{\mathcal{F}_{\mathrm{tf}}} \tilde{\varepsilon}}}{1 - \gamma} + \frac{V_{\max}\sqrt{C_{\mathcal{F}_{\mathrm{tf}}}}}{(1-\gamma)\sqrt{n}} \sqrt{mL^2 d^2 \log \frac{mdL\bar{B}n}{V_{\max}} + \log \frac{2\mathcal{N}(\Pi, 1/n, d_\infty)}{\delta}} \right),$$

*where $d = d_{\mathcal{S}} + d_{\mathcal{A}}$, $\tilde{\varepsilon} = \varepsilon_{\mathcal{F}} + \varepsilon_{\mathcal{F},\mathcal{F}}$, and $\bar{B}$ is defined in Proposition 2.*

Theorem 3 shows that the upper bound of the suboptimality gap does not scale with the number of agents $N$, which demonstrates that the proposed model-free algorithm breaks the curse of many agents. We note that the model-free offline/batch MARL with homogeneous agents has been studied in [8] and [22], and the suboptimality upper bounds in [8, Theorem 1] and [22, Theorem 4.1] are also independent of $N$. However, these works adopt the mean-field approximation of the original MDP, in which the influence of all the other agents on a specific agent is only coarsely considered through the distribution of the state. The approximation error between the action-value function of the mean-field MDP and that of the original MDP is not analyzed therein. Thus, the independence of $N$ in their works comes with the cost of the poor relational reasoning ability and the unspecified approximation error. In contrast, we analyze the suboptimality gap of the learned policy in the original MDP, and the interaction among agents is captured by the transformer network.

### 4.2 Pessimistic Model-based Offline Reinforcement Learning

In this subsection, we present the model-based algorithm, where we adopt the transformer to estimate the system dynamics and learn the policy based on such an estimate.

#### 4.2.1 Neural Nonlinear Regulator

In this section, we consider the Neural Nonlinear Regulator (NNR), in which we use the transformer to estimate the system dynamics. The ground truth transition $P^*(\bar{S}' \,|\, \bar{S}, \bar{A})$ is defined as $\bar{S}' = F^*(\bar{S}, \bar{A}) + \bar{\varepsilon}$, where $F^*$ is a nonlinear function, $\bar{\varepsilon} = [\varepsilon_1, \ldots, \varepsilon_N]^\top$ is the noise, and $\varepsilon_i \sim \mathcal{N}(0, \sigma^2 I_{d \times d})$ for $i \in [N]$ are independent random vectors. We note that the function $F^*$ and the transition kernel $P^*$ are equivalent, and we denote the transition kernel corresponding to the function $F$ as $P_F$. Since the transition kernel $P^*(\bar{S}' \,|\, \bar{S}, \bar{A})$ is permutation invariant, $F^*$ should be permutation equivariant, i.e., $F^*(\psi(\bar{S}), \psi(\bar{A})) = \psi(F^*(\bar{S}, \bar{A}))$ for all row-wise permutation functions $\psi(\cdot)$.

We take $X = [\bar{S}, \bar{A}] \in \mathbb{R}^{N \times d}$ as the input of the network and adopt a similar network structure as the transformer specified in Section 3.2. However, to predict the next state instead of the action-value function with the transformer, we remove the average aggregation module in the final layer of the structure in Section 3.2. Please refer to Appendix B for the formal definition. The permutation equivariance of the proposed transformer structure can be easily proved with the permutation equivariance of the self-attention mechanism. We consider the transformer function class with bounded parameters, which is defined as

$$\mathcal{M}_{\mathrm{tf}}(B') = \Big\{ F_{\mathrm{tf}}(X; W_{QK}^{1:L}, W_V^{1:L}, a^{1:L}, b^{1:L}) \,\Big|\, |a_{kj}^{(i)}| < B_a, \|b_{kj}^{(i)}\|_2 < B_b,$$

$$\big\| W_{QK}^{(i)\top} \big\|_{\mathrm{F}} < B_{QK}, \big\| W_V^{(i)\top} \big\|_{\mathrm{F}} < B_V \text{ for } i \in [L], j \in [m], k \in [d] \Big\},$$

where $B' = [B_a, B_b, B_{QK}, B_V]$ is the vector of parameters of the function class. We denote $\mathcal{M}_{\mathrm{tf}}(B')$ as $\mathcal{M}_{\mathrm{tf}}$ when the parameters are clear from the context.

#### 4.2.2 Algorithm

Given the offline dataset $\mathcal{D} = \{(\bar{S}_i, \bar{A}_i, r_i, \bar{S}_i')\}_{i=1}^n$, we first derive the MLE of the system dynamics. Next, we learn the optimal policy according to the confidence region of the dynamics that are

constructed around the MLE. The term "pessimism" is reflected in the procedure that we choose the system dynamics that induce the *smallest* value function, i.e.,

$$\hat{F}_{\mathrm{MLE}} = \operatorname*{argmin}_{F \in \mathcal{M}_{\mathrm{tf}}} \frac{1}{n} \sum_{i=1}^{n} \left\| \bar{S}_i' - F(\bar{S}_i, \bar{A}_i) \right\|_{\mathrm{F}}^2 \quad \text{and} \quad \hat{\pi} = \operatorname*{argmax}_{\pi \in \Pi} \min_{F \in \mathcal{M}_{\mathrm{MLE}}(\zeta)} V_{P_F}^{\pi}(\bar{S}_0), \quad (3)$$

where $\mathcal{M}_{\mathrm{MLE}}(\zeta) = \{F \in \mathcal{M}_{\mathrm{tf}}(B') \,|\, 1/n \cdot \sum_{i=1}^{n} \mathrm{TV}(P_F(\cdot\,|\,\bar{S}_i, \bar{A}_i), \hat{P}_{\mathrm{MLE}}(\cdot\,|\,\bar{S}_i, \bar{A}_i))^2 \le \zeta\}$ is the confidence region, which has a closed-form expression in terms of the difference between $F$ and $\hat{F}_{\mathrm{MLE}}$ as stated in in Appendix C. The transition kernel induced by $\hat{F}_{\mathrm{MLE}}$ is denoted as $\hat{P}_{\mathrm{MLE}}$. The parameter $\zeta$ is used to measure the tolerance of estimation error of the system dynamics, and it is set according to the parameters of $\mathcal{M}_{\mathrm{tf}}(B')$ such that $F^*$ belongs to $\mathcal{M}_{\mathrm{MLE}}(\zeta)$ with high probability.

Similar to the model-free algorithm, the model-based algorithm specified in Eqn. (3) does not require global coverage. Instead, the mismatch between the distribution induced by the optimal policy $d_{P^*}^{\pi^*}$ and the sampling distribution $\nu$ is captured by the constant

$$C_{\mathcal{M}_{\mathrm{tf}}} = \max_{F \in \mathcal{M}_{\mathrm{tf}}} \mathbb{E}_{d_{P^*}^{\pi^*}} \left[ \mathrm{TV}\big(P_F(\cdot\,|\,\bar{S}, \bar{A}), P^*(\cdot\,|\,\bar{S}, \bar{A})\big)^2 \right] \Big/ \mathbb{E}_{\nu} \left[ \mathrm{TV}\big(P_F(\cdot\,|\,\bar{S}, \bar{A}), P^*(\cdot\,|\,\bar{S}, \bar{A})\big)^2 \right]. \quad (4)$$

We note that $C_{\mathcal{M}_{\mathrm{tf}}} \le \max_{(\bar{S}, \bar{A}) \in \bar{\mathcal{S}} \times \bar{\mathcal{A}}} d_{P^*}^{\pi^*}(\bar{S}, \bar{A})/\nu(\bar{S}, \bar{A})$, so the suboptimality bound involving $C_{\mathcal{P}_{\mathcal{F}_{\mathrm{tf}}}}$ in Theorem 4 is tighter than the bound requiring global convergence. Similar coefficients also appear in many existing works such as [42] and [20].

### 4.2.3  Analysis of the Maximum Likelihood Estimate

Every $F \in \mathcal{M}_{\mathrm{MLE}}(\zeta)$ is near to the MLE in the total variation sense and thus well approximates the ground truth system dynamics. Therefore, to derive an upper bound of the suboptimality gap of the learned policy, we first analyze the convergence rate of the MLE $\hat{P}_{\mathrm{MLE}}$ to $P^*$.

**Proposition 3.** *Let $\tilde{B} = B_V B_{QK} B_a B_b$. For the maximum likelihood estimate $\hat{P}_{\mathrm{MLE}}$ in Eqn. (3), the following inequality holds with probability at least $1 - \delta$,*

$$\mathbb{E}_{\nu} \left[ \mathrm{TV}\big(P^*(\cdot\,|\,\bar{S}, \bar{A}), \hat{P}_{\mathrm{MLE}}(\cdot\,|\,\bar{S}, \bar{A})\big)^2 \right] \le O\left( \frac{1}{n} m L^2 d^2 \log\big(NLmd\tilde{B}n\big) + \frac{1}{n} \log \frac{1}{\delta} \right).$$

We define $e'(\mathcal{M}_{\mathrm{tf}}, n)$ to be $n$ times the total variation bound. Proposition 3 shows that the total variation estimation error is polynomial in the depth of the neural network $L$. However, different from the model-free RL results in Section 4.1, the estimation error of MLE $\hat{P}_{\mathrm{MLE}}$ is logarithmic in the number of agents $N$. We note that this logarithm dependency on $N$ comes from the fact that $\mathrm{TV}(P^*(\cdot\,|\,\bar{S}, \bar{A}), \hat{P}_{\mathrm{MLE}}(\cdot\,|\,\bar{S}, \bar{A}))$ measures the distance between two transition kernels that involves the states of $N$ agents, different from the scalar estimate of the value function in Section 4.1. To prove the result, we adopt a PAC-Bayesian framework to analyze the convergence rate of MLE, which is inspired by the analysis of density estimation [43]; more details are presented in Appendix J.

### 4.2.4  Bound on the Suboptimality Gap

To analyze the error of the learned model, we make the following realizability assumption.

**Assumption 3.** *The nominal system dynamics belongs to the function class $\mathcal{M}_{\mathrm{tf}}$, i.e., $F^* \in \mathcal{M}_{\mathrm{tf}}(B')$.*

In addition, we require that the mismatch between the sampling distribution and the visitation distribution of the optimal policy is bounded.

**Assumption 4.** *For the sampling distribution $\nu$, the coefficient $C_{\mathcal{M}_{\mathrm{tf}}}$ defined in (4) is finite.*

We note that these two assumptions are also made in many existing works, e.g., [20, 21].

**Theorem 4.** *If Assumptions 3 and 4 hold, and we take $\zeta = c_1 e'(\mathcal{M}_{\mathrm{tf}}, n)/n$ for some constant $c_1 > 0$, then with probability at least $1 - \delta$, the suboptimality gap of the policy learned in the algorithm in Eqn. (3) is upper bounded as*

$$V_{P^*}^{\pi^*}(\bar{S}_0) - V_{P^*}^{\hat{\pi}}(\bar{S}_0) \le O\left( \frac{V_{\max}}{(1-\gamma)^2} \sqrt{C_{\mathcal{M}_{\mathrm{tf}}} \left( \frac{1}{n} m L^2 d^2 \log\big(NLmd\tilde{B}n\big) + \frac{1}{n} \log \frac{1}{\delta} \right)} \right),$$

*where $d = d_{\mathcal{S}} + d_{\mathcal{A}}$, and $\tilde{B}$ is defined in Proposition 3.*

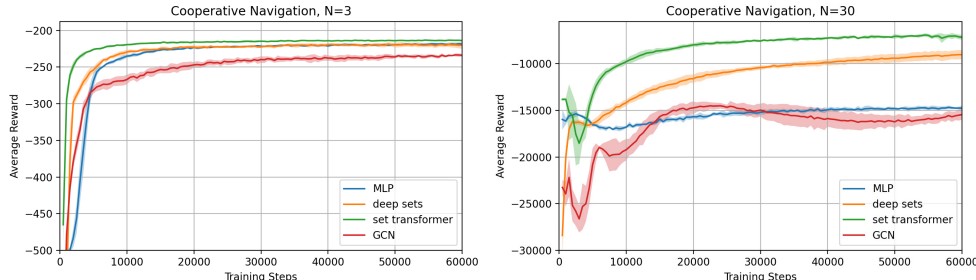

Figure 3: Average rewards of model-free RL algorithms with their standard deviations for $N = 3, 30$.

Theorem 4 presents an upper bound on the suboptimality gap of the offline model-based RL with the transformer approximators. The suboptimality gap depends on the number of agents only as $O(\sqrt{\log N})$, which shows that the proposed model-based MARL algorithm mitigates the curse of many agents. This weak dependence on $N$ originates from measuring the distance between two system dynamics of $N$ agents in the learning of the dynamics. To the best of our knowledge, there is no prior work on analyzing the model-based algorithm for the homogeneous MARL, even from the mean-field approximation perspective. The proof of Theorem 4 leverages novel analysis of the MLE in Proposition 3. For more details, please refer to Appendix H.

## 5 Experimental Results

We evaluate the performance of the algorithms on the Multiple Particle Environment (MPE) [44, 45]. We focus on the *cooperative navigation* task, where $N$ agents move cooperatively to cover $L$ landmarks in an environment. Given the positions of the $N$ agents $x_i \in \mathbb{R}^2$ (for $i \in [N]$) and the positions of the $L$ landmarks $y_j \in \mathbb{R}^2$ (for $j \in [L]$), the agents receive reward $r = -\sum_{j=1}^{L} \min_{i \in [N]} \|y_j - x_i\|_2$. This reward encourages the agents to move closer to the landmarks. We set the number of agents as $N = 3, 6, 15, 30$ and the number of landmarks as $L = N$. Here, we only present the result for $N = 3, 30$. Please refer to Appendix O for more numerical results. To collect an offline dataset, we learn a policy in the online setting. Then the offline dataset is collected from the induced stationary distribution of such a policy. We use MLP, deep sets, Graph Convolutional Network (GCN) [46], and set transformer to estimate the value function. We note that the deep sets, GCN, and set transformer are permutation invariant functions. For the implementation details, please refer to Appendix O.

Figure 3 shows that the performances of the MLP and deep sets are worse than that of the set transformer. This is due to the poor relational reasoning abilities of MLP and deep sets, which corroborates Theorem 1. Figure 3 indicates that when the number of agents $N$ increases, the superiority of the algorithm with set transformer becomes more pronounced, which is strongly aligned with our theoretical result in Theorem 3.

## 6 Concluding remarks

In view of the tremendous empirical successes of cooperative MARL with permutation invariant agents, it is imperative to develop a firm theoretical understanding of this MARL problem because it will inspire the design of even more efficient algorithms. In this work, we design and analyze algorithms that break the curse of many agents and, at the same time, implement efficient relational reasoning. Our algorithms and analyses serve as a first step towards developing provably efficient MARL algorithms with permutation invariant approximators. We leave the extension of our results of the transformer to *general permutation invariant approximators* as future works.

## Acknowledgments and Disclosure of Funding

Fengzhuo Zhang and Vincent Tan acknowledge funding by the National Research Foundation Singapore and DSO National Laboratories under the AI Singapore Programme (AISG Award No: AISG2-RP-2020-018) and by Singapore Ministry of Education (MOE) AcRF Tier 1 Grants (A-0009042-01-00 and A-8000189-01-00). Zhaoran Wang acknowledges the National Science Foundation (Awards 2048075, 2008827, 2015568, 1934931), Simons Institute (Theory of Reinforcement Learning), Amazon, J. P. Morgan, and Two Sigma for their support.

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
