# OpenReview forum: "Relational Reasoning via Set Transformers: Provable Efficiency and Applications to MARL"
_NeurIPS.cc/2022/Conference — NeurIPS 2022 Accept_

### Official Review · Reviewer_eLHD · 2022-07-09

**Rating:** 6
**Confidence:** 3
**Soundness:** 3 good
**Presentation:** 3 good
**Contribution:** 2 fair

**Summary:**

This paper concerns the theoretical understanding and relational reasoning of permutation invariant agents framework in MARL. It proposes offline MARL with the transformer and analyze the error bound.

**Questions:**

- The paper compares relational reasoning between self-attention and MLP through the width of NN, what about the parameters and computation?
- Could the method execution decentralized?

**Limitations:**

- The environment is simple and can't empirically demonstrate the performance of the method.
- The novelty is not enough. The paper extends single-agent offline RL and utilize set transformers as neural network structure.

**Strengths And Weaknesses:**

- It utilizes self-attention mechanism that is widely used in cv and nlp to model relational reasoning between agents
- It proposes both model-free and model-based offline MARL
- It theoretically prove the gap does not scale with the number of agents and the proof is complete

---

> ### Author Response · Authors · 2022-08-02
> **Response to Reviewer eLHD: Part 1**
>
> We thank the reviewer for the valuable feedback. We appreciate the suggestions on writing and will revise the paper accordingly. We address the major comments in the following.
>
>
> **Comparison of parameters and computation**:
> Parameters: The number of parameters of the deep set function class (considered in Theorem 1) is linear in the width of the neural network and the dimension in each channel. Hence, the number of parameters of the deep set is $O(d\exp(cd))$ for some constant $c>0$. In contrast, the number of parameters of the transformer considered in Theorem 1 is $O(d)$. Thus, using transformers, we have **exponentially fewer parameters**.
> Computation: For the MLP and the transformer function class considered in Theorem 1, the computational complexity of evaluating the MLP is $O(Nd\exp(c^{\prime}d))$ for some constant $c^{\prime}>0$, and the computational complexity of evaluating the transformer is $O(N^{2}d)$. However, we note that the computational complexity of the transformer can be reduced to $O(Nd\log N)$ by appropriate reformulations and approximations; see the techniques—Performer [1], Linformer [2] and Reformer [3].
>
> **Decentralized execution**: We consider general policy classes in this work. After setting the policy set $\Pi$ to be the set of policies that can be expressed as the product of the policy of each agent, each agent can take actions according to his/her own policy in **a decentralized way**. In this case, our algorithms can be viewed as Centralized Training Decentralized Execution (CTDE) algorithms [4].
>
> **Simulation environment**: We verify our theoretical results on the cooperative navigation task in the Multiple Particle Environment (MPE). This environment is widely used in [5,6,7]. In the original submission, we presented the simulation results for the case where the number of agents $N$ is set to 15 in Appendix O. In Appendix O of our updated version of the supplementary materials, we augment the simulation results to elucidate the effect on the number of agents. We present the table of the **final rewards** with different neural networks and different number of agents $N$ here. The rewards reported are the averaged rewards over the last 10 training episodes. When the environment becomes more complex, i.e., there are more agents, the superiority of adopting the transformer to estimate the action-value function is more evident. These experimental results corroborate our theoretical results in Theorems 1 and 2. Besides our simulation results, the superiority of the transformer in RL problems has been empirically demonstrated in different environments in other works [8,9,10].
>
> |     &emsp; &ensp; &nbsp; | &emsp; MLP &emsp;&nbsp;|Deep sets| Set transformer| &emsp; GCN  &emsp;&nbsp;|
> | &nbsp; -----&nbsp; |&ensp;-----------&emsp;|&nbsp;-------------&thinsp;|&emsp;---------------- &thinsp;&emsp;|&emsp;----------&emsp;|
> | N=3  &nbsp; | -218.66   &ensp;&ensp;| -219.97&thinsp; &ensp;| &emsp;-213.66 &ensp;&emsp;&emsp;| -234.56   &emsp; |
> | N=6  &nbsp; | -4096.53 &nbsp; | -3476.76   | &emsp;-2936.54 &emsp;&emsp;| -3731.03&ensp;&thinsp; |
> | N=15&nbsp;| -3482.63  &ensp;| -1466.11   | &emsp;-1390.90 &emsp;&emsp;| -1831.01&ensp;&thinsp; |
> | N=30  | -14751.74  | -9112.87   | &emsp;-7096.56 &emsp;&emsp;| -15728.25  |

---

> > ### Author Response · Authors · 2022-08-02
> > **Response to Reviewer eLHD: Part 2**
> >
> > **Originality:** We would like to take this opportunity to reiterate the originality of our work here.
> >
> > First, specifying an efficient function class in MARL is an important problem. Unlike the single-agent RL problem, the design of the function class is among the  most challenging questions in the MARL problems due its effect on the **complex interactions** among agents and **the curse of many agents**.
> > Besides, the **generalization ability** and the **expressiveness** of the function class heavily impact policy evaluation and policy optimization, which are two core subroutines in all MARL algorithms.
> > Thus, it seems unclear which function class is sufficiently expressive to capture the complicated interactions and mitigate the curse of many agents.
> > Given the tremendous empirical success of adopting transformer in MARL problems [8,9], we **justify from a theoretical perspective** the use of the transformer by demonstrating its **superiority of relational reasoning** in Theorem 1 and the **advantage generalization-wise** in Proposition 2. In particular, Proposition 2 shows that the transformer class does not suffer from the curse of many agent by enjoying a generalization bound that is indepdent on $N$.
> >
> > Second, the **analysis of the MARL algorithm with the efficient transformer function class is difficult**, and we make some progress in our paper. The analysis of the model-free and model-based MARL algorithm requires a tight generalization bound of the transformer and the uncertainty quantification of the system dynamics with the transformer. We provide an improved generalization bound of the transformer in Proposition 2 and a quantification of the uncertainty resulting from the MLE in Proposition 3. Our generalization bound in Proposition 2 can also be **used in conjunction with the analyses of other RL algorithms with the transformer approximator**, such as the actor-critic RL algorithm. The reviewer is partially correct that we extend single agent RL and utilize the set transformer; however, the **analysis** of this combination, which is the main contribution of the present work, is, we believe, highly non-trivial.
> >
> > [1] K. Choromanski, et al., Rethinking attention with performers. International conference on learning representations, 2021.
> > [2] S. Wang, B. Z. Li, M. Khabsa, H. Fang, and H. Ma. Linformer: Self-attention with linear complexity. arXiv preprint arXiv:2006.04768, 2020.
> > [3] N. Kitaev, L. Kaiser, and A. Levskaya, Reformer: The efficient transformer. International conference on learning representations, 2020.
> > [4] X. Lyu, Y. Xiao, B. Daley, and C. Amato. Contrasting centralized and decentralized critics in multi-agent reinforcement learning. In Proceedings of the 20th International Conference on Autonomous Agents and Multi-Agent Systems, 2021.
> > [5] R. Lowe, Y. I. Wu, A. Tamar, J. Harb, OpenAI Pieter A., and I. Mordatch. Multi-agent actor-critic for mixed cooperative-competitive environments. Advances in neural information processing systems, 30, 2017.
> > [6] I. Mordatch and P. Abbeel. Emergence of grounded compositional language in multi-agent populations. In Proceedings of the AAAI Conference on Artificial Intelligence, volume 32, 2018.
> > [7] I. Liu, R. A. Yeh, and A. G. Schwing. PIC: permutation invariant critic for multi-agent deep reinforcement learning. In Conference on Robot Learning, pages 590–602. PMLR, 2020.
> > [8] T. Zhou et al. Cooperative multi-agent transfer learning with level-adaptive credit assignment, arXiv preprint arXiv:2106.00517, 2021.
> > [9] M. Wen, J. G. Kuba, R. Lin, W. Zhang, Y. Wen, J. Wang, and Y. Yang. Multi-Agent Reinforcement Learning is a Sequence Modeling Problem. arXiv preprint arXiv:2205.14953, 2022.
> > [10] K. Wang, H. Zhao, X. Luo, K. Ren, W. Zhang, and D. Li. Bootstrapped Transformer for Offline Reinforcement Learning. arXiv preprint arXiv:2206.08569, 2022.

---

> > ### Comment · Reviewer_eLHD · 2022-08-03
> > **Response**
> >
> > * Have you do some experiment on StartCraft? Only MPE may be not enough to support the proposed approach.
> > * What's the ralation between multi-agent decision transformer([1],[2]) and this work? Since they all use transformer but from the different perspective
> > [1] Offline pre-trained multi-agent decision transformer: One big sequence model conquers all starcraftii tasks.
> > [2] Multi-Agent Reinforcement Learning is a Sequence Modeling Problem

---

> > > ### Author Response · Authors · 2022-08-04
> > > **Re: Response**
> > >
> > > We thank the reviewer for raising these questions concerning experimental validations.
> > >
> > > **Starcraft environment**: These are certainly valid considerations. Nevertheless, the main contribution of our work is in the **theoretical analysis** that corroborates the superiority of the transformer via its application to MARL. In contrast, most existing works on MARL with transformers do not present theoretical analyses of the advantages of transformers that have been observed experimentally. Our work does not focus on improving the SOTA; this endeavor is orthogonal to the main objective of our paper which is to place the advantages of transformers in MARL on a firm **theoretical** footing.
> > >
> > > Our goal is to theoretically validate that the transformer function class is an appropriate function class for offline MARL, and we have validated the superiority of the transformer in the MPE environment in Appendix O, which strongly corroborates our theoretical results. Although running further experiments in various MARL environments, like StarCraft, and trying to achieve SOTA is definitely interesting, but this is beyond the scope of this work. We seek the reviewer’s kind understanding on this matter.
> > >
> > > **Relation with [1] and [2]**: We discuss the relationship between our work and [1,2] from perspectives of main contributions, algorithms, and the underlying connections.
> > >
> > > First, the main contribution of our work is to provide a **theoretical analysis** of the superiority of the transformer via its application to MARL. In contrast, Meng et al. [1] and Wen et al. [2] manage to improve the **experimental performance** of some tasks by optimizing the transformer structure for MARL problems.
> > >
> > > Second, we consider the **offline**  MARL algorithm with transformer approximators in our work.  However, Meng et al. [1] focus on improving the performance of **downstream tasks** through pre-training a generalized policy, and Wen et al. [2] focus on the **online** MARL with Multi-Agent Transformer (MAT). Our theoretical results potentially apply to the settings in [1,2], but such generalization requires an understanding of the mechanism of pre-training (for setting of [1]) and Eluder dimension of neural networks (for online RL), which are left as future directions.
> > >
> > > Finally, we highlight that our results provide a firm footing for the **theoretical analysis** of the algorithms in [1] and [2]. For example, Wen et al. [2] adopt an encoder to latent representations of the observations, which is training by minimizing the empirical Bellman error. Our generalization bound of the transformers in Proposition 2 can be adopted in the analysis of generalization ability of the learned encoder, but as mentioned, other techniques (such as the analysis of the Eluder dimension of neural networks) need to be developed.
> > >
> > > [1] L, Meng et al. Offline pre-trained multi-agent decision transformer: One big sequence model conquers all starcraftII tasks. arXiv preprint arXiv:2112.02845, 2021.
> > > [2] M, Wen et al. Multi-Agent Reinforcement Learning is a Sequence Modeling Problem. arXiv preprint arXiv:2205.14953, 2022.

---

### Official Review · Reviewer_TVHt · 2022-07-12

**Rating:** 7
**Confidence:** 1
**Soundness:** 4 excellent
**Presentation:** 1 poor
**Contribution:** 3 good

**Summary:**

This paper presents theoretical analysis on the use of transformers for offline multi-agent reinforcement learning. The main contributions are: i) a proof that approximating the "relational reasoning" of set transformers using feed-forward neural nets requires exponential width, ii) model-free and model-based algorithms for offline MARL using transformers and pessimistic policies, which minimize the effect of distribution shift, and iii) suboptimality gaps for the proposed algorithms showing that they scale well with the number of agents.

**Questions:**

No questions.

**Limitations:**

I didn't see any discussion on limitations in this paper, which in fact doesn't have a final discussion/conclusion section.

**Strengths And Weaknesses:**

Unfortunately, I found this paper very hard to understand, even after spending several hours on it and multiple reads. Not only it is theory intensive, but it often introduces ideas and terminology too suddenly and with very sparse explanations. Since this is not my area of expertise, I will opt for assuming that the math and derivations are correct, in which case I think this paper is probably a good contribution to the conference. The topic is clearly relevant, and the use of transformers is gaining prominence in in reinforcement learning, so analysis such as the one presented in this paper are of great interest. But again, I must qualify this opinion with the caveat that I'm taking the results offered at face value.

---

> ### Author Response · Authors · 2022-08-02
> **Response to Reviewer TVHt**
>
> We thank the reviewer for the valuable feedback. We appreciate the suggestions on writing and will revise our paper accordingly, should it be accepted. We address the major comments in the following.
>
> **Limitation discussion**: In this paper, we corroborate, from a theoretical perspective, the efficacy of the transformer on the offline MARL problem. We note that the generalization bound of the transformer in Proposition 2 can also be conveniently applied to (or plugged into) other RL settings. However, the detailed analysis of the other RL settings with transformer approximators is not presented in this paper due to the space constraints. We leave this for future work. We will add a conclusion section should the paper be accepted since we will have an additional page.

---

### Official Review · Reviewer_gWc2 · 2022-07-12

**Rating:** 4
**Confidence:** 3
**Soundness:** 3 good
**Presentation:** 2 fair
**Contribution:** 2 fair

**Summary:**

The paper tackles the problem of efficient offline RL in the multi-agent setting (MARL) that typically suffers from the curse of dimensionality as the number of agents grows. They argue that transformers are ideally suited for estimating the RL components (value functions/dynamics models) and are able to implement efficient relational reasoning between agents.
They combine model-free and model-based RL algorithms with transformers as function approximators for the value function and dynamics model respectively, and present theoretical results for the generalisation and suboptimality gaps for the resulting algorithms.

**Questions:**

See comments above regarding assumptions and applicability in more realistic settings. I would be also curious to know if the authors can say something about the online MARL setting when using transformers?

Given the topic of the paper (how a neural network architecture can give rise to efficient algorithms), it could greatly benefit from an experimental demonstration of the theoretical results.

The conclusions discussion should be part of the main text, not supplementary.



**Limitations:**

As mentioned before, the authors should elaborate on the assumptions necessary for the results and what we can expect in practice when they don't hold.

**Strengths And Weaknesses:**

The papers main contribution is the theoretical analysis of MARL algorithms when using transformers for function approximation, showing that they can get significantly tighter bounds on the generalisation error e.g. for the model-free algorithm the error bound becomes independent of the number of agents.

I am not familiar with details of prior work on offline MARL and the presentation of the paper made it often difficult to judge the significance and the originality of some of the contributions. E.g. "we design offline model-free and model-based RL algorithms with the transformer approximators" It seemed to me that the authors modified existing RL algorithms by simply replacing the function approximator used with transformers?

I would have appreciated more discussion on the assumptions required for the results (to be able to identify which ones are the strongest), instead of simply referring to other works that make similar assumptions. E.g. the IID assumption (instead of sequential) on the offline data.

The paper is missing a discussion and/or experimental demonstration of how much of the favourable scaling properties would carry over to more realistic settings which would make the significance of the results a lot more clear.

---

> ### Author Response · Authors · 2022-08-02
> **Response to Reviewer gWc2: Part 1**
>
> We thank the reviewer for the valuable feedback. We appreciate the suggestions on experiments and will revise accordingly in the revision. We address the major concerns in the following.
>
> **Originality:** We would like to take this opportunity to reiterate the originality of our work here.
>
> First, specifying an efficient function class in MARL is an important problem. Unlike the single-agent RL problem, the design of the function class is among the  most challenging questions in the MARL problems due its effect on the **complex interactions** among agents and **the curse of many agents**.
> Besides, the **generalization ability** and the **expressiveness** of the function class heavily impact policy evaluation and policy optimization, which are two core subroutines in all MARL algorithms.
>
> Thus, it seems unclear which function class is sufficiently expressive to capture the complicated interactions and mitigate the curse of many agents.
>
> Given the tremendous empirical success of adopting transformer in MARL problems [1,2], we **justify from a theoretical perspective** the use of the transformer by demonstrating its **superiority of relational reasoning** in Theorem 1 and the **advantage generalization-wise** in Proposition 2. In particular, Proposition 2 shows that the transformer class does not suffer from the curse of many agent by enjoying a generalization bound that is indepdent on $N$.
>
> Second, the **analysis of the MARL algorithm with the efficient transformer function class is difficult**, and we make some progress in our paper. The analyses of the model-free and model-based MARL algorithms require  tight generalization bounds of the transformer and the uncertainty quantification of the system dynamics with the transformer. We provide an improved generalization bound of the transformer in Proposition 2 and a quantification of the uncertainty resulting from the maximum likelihood estimate (MLE) in Proposition 3. Our generalization bound in Proposition 2 can also be **used in conjunction with the analyses of other RL algorithms with the transformer**, such as the actor-critic RL algorithm.
>
> **Explanation of assumptions:**
> We would like to take this opportunity here to explain the necessity of the 5 assumptions we made in the paper.
>
> * IID assumption: We only adopt the i.i.d. sampling procedure in the main paper in order to simplify the definition and the presentation of our theoretical results. Our proof techniques for improving the generalization bound of transformers can be plugged-in the proof of concentration results under other settings. In Appendix N.2 of our updated version of supplementary materials, we extend our Proposition 2 and Theorem 2 to the (more common) **sequential** sampling setting, in which the state-action pair evolves according to a Markov chain. We assume that the offline dataset $\mathcal{D}^{\prime}$ is formed by implementing a policy $\pi_{0}$ on the underlying MDP. Denote the stationary distribution and the absolute spectral gap of the Markov chain of state-action pair induced by the policy $\pi_{0}$ as $q_{P^{*}}^{\pi_{0}}=\mu^{\prime}$ and $1-\lambda$, respectively. Then the generalization error bound of transformers can be derived as follows.
> **Proposition 23**
> 	Consider the dataset $\mathcal{D}^{\prime}$ collected by implementing a policy $\pi_{0}$. Let $\bar{B}=B_{V}B_{QK}B_{a}B_{b}B_{w}$.  For all $f,\tilde{f}\in\mathcal{F}_{\mathrm{tf}}(B)$ and all policies $\pi\in\Pi$, with probability at least $1-\delta$, we have
>
> \begin{align}
> \qquad\Big|\mathbb{E}_{\mu^{\prime}}\Big[\big(f(\bar{S},\bar{A})-\mathcal{T}^{\pi}\tilde{f}(\bar{S},\bar{A})\big)^{2}\Big]-\mathcal{L}^{\prime}(f,\tilde{f},\pi;\mathcal{D}^{\prime})+\mathcal{L}^{\prime}(\mathcal{T}^{\pi}\tilde{f},\tilde{f},\pi;\mathcal{D}^{\prime})\Big|
> \end{align}
>
> \begin{align}
> \qquad\leq\frac{C+(2-C)\lambda}{2}\mathbb{E}_{\mu^{\prime}}\Big[\big(f(\bar{S},\bar{A})-\mathcal{T}^{\pi}\tilde{f}(\bar{S},\bar{A})\big)^{2}\Big]
> \end{align}
>
> \begin{align}
> \qquad+O\bigg(\frac{V_{\max}^{2}}{(1-\lambda)n}\biggl[mL^{2}d^{2}\log\frac{mdL\bar{B} n}{V_{\max}} +\log\frac{N(q_{0}\pi_{0},q_{P^{*}}^{\pi_{0}})\mathcal{N}(\Pi,1/n,d_{\infty})}{\delta}\biggr]\bigg)
> \end{align}
> where $1-\lambda$ is the absolute spectral gap of the Markov chain of $(\bar{S},\bar{A})$  induced by the policy $\pi_{0}$, and $0<C<e^{1/10}$ is an absolute constant.
> For ease of notation, we define $\tilde{e}(\mathcal{F}, \Pi,\pi_{0},\delta,n)$ to be $(1-\lambda)n$ times the second term of the generalization error bound, where $\mathcal{F}$ is the transformer function class. We note that Proposition 23 is a generalization of Proposition 2. When the dataset $\mathcal{D}$ consists of i.i.d. samples according to $\nu$, the dataset $\mathcal{D}$ can be treated as a Markov chain with $\lambda=0$, and $N(\nu,\nu)=1$. In this case, our result in Proposition 23 particularizes to the result in  Proposition 2 up to a constant.

---

> > ### Author Response · Authors · 2022-08-02
> > **Response to Reviewer gWc2: Part 2**
> >
> > Under this sequential sampling model, the mismatch between the  distribution induced by the optimal policy and the stationary distribution $q_{P^{*}}^{\pi_{0}}$ is captured by  $C_{\mathcal{F}}^{\prime}(\pi_{0})$, which is similarly defined as $C_{\mathcal{F}_{\text{tf}}}$ in Eqn. (2). Formal definition is in Appendix N.2 in our updated supplementary materials. Then The suboptimality bound of the learned policy is derived as follows.
> >
> > **Theorem 5**
> > If $C_{\mathcal{F}}^{\prime}(\pi_{0})$ is finite and the transformer function class is approximate realizable and completer with parameters $\varepsilon_{\mathcal{F}}^{\prime}$ and $\varepsilon_{\mathcal{F},\mathcal{F}}^{\prime}$, and we take $\varepsilon=[2+C+(2-C)\lambda]\varepsilon_{\mathcal{F}}^{\prime}/2 +2\tilde{e}(\mathcal{F},\Pi,\pi_{0},\delta,n)/[(1-\lambda)n]$ to define the confidence region of the action-value function, then with probability at least $1-\delta$, the suboptimality gap of the policy derived in our proposed model-free RL algorithm with sequentially sampled dataset $\mathcal{D}^{\prime}$ in is upper bounded as (here we slightly change the notation of the value function of a policy due to the complie issues in Openreview. Please see Appendix N.2 of our updated supplementary materials for the formal statement.)
> > \begin{align}
> > V(\pi^{*})-V(\hat{\pi})\leq O\Bigg(\sqrt{\frac{C_{\mathcal{F}}^{\prime}(\pi_{0})\tilde{\varepsilon}}{(1-\gamma)^{2}(1-\lambda)}}+\frac{V_{\max}\sqrt{C_{\mathcal{F}}^{\prime}(\pi_{0})  }}{(1-\gamma)(1-\lambda)\sqrt{n} }\cdot
> > \end{align}
> >
> > \begin{align}
> > \sqrt{mL^{2}d^{2}\log\frac{mdL\bar{B}n}{V_{\max}}+\log \frac{2N(q_{0}\pi_{0},q_{P^{*}}^{\pi_{0}})\mathcal{N}(\Pi,1/n,d_{\infty})}{\delta} }\Bigg)
> > \end{align}
> > where $d=d_{\mathcal{S}}+d_{\mathcal{A}}$, $\tilde{\varepsilon} = \varepsilon_{\mathcal{F}}^{\prime} + \varepsilon_{\mathcal{F},\mathcal{F}}^{\prime}$, $\bar{B}$ is defined in Proposition 2, $0<C<e^{1/10}$ is an absolute constant, and $1-\lambda$ is the absolute spectral gap of the Markov chain of $(\bar{S},\bar{A})$ induced by the policy $\pi_{0}$.
> > Theorem 5 is a generalization of Theorem 2. Sampling in an i.i.d.  manner according to $\nu$ can be regarded as a Markov chain with $\lambda=0$, and $N(\nu,\nu)=1$. In this case, our result in Theorem 5 particularizes to the result in  Theorem 2.
> >
> > * Assumption 1: This assumption concerns the **approximate realizability** and **completeness** of transformers. The parameter $\varepsilon_{\mathcal{F}}$ quantifies the minimal distance between the action-value function and the function class, and $\varepsilon_{\mathcal{F},\mathcal{F}}$ quantifies the Bellman error with respect to any policy. This assumption ensures that we can derive a meaningful estimate of the action-value function, and it holds when the function class has strong approximation capabilities. We note that such an assumption has been widely adopted, e.g., in [3,4,5]. For the transformer function class, Yun et al. [6] show that it is a **universal approximator** of the continuous function with a compact domain with respect to the $l_{p}$-norm [6]. Increasing the width and the depth of the transformer helps to approximate other kinds of functions, but this will increase the generalization error in Proposition 2 at the same time.
> >
> > * Assumption 2: This assumption essentially characterizes the **distributional shift** between the visitation measure induced by the optimal policy and that induced by the behavior policy.  More concretely, this assumption states that the Bellman error with respect to the optimal policy computed along the trajectory induced by the optimal policy can be bounded by that induced by the sampling distribution $\nu$ up to a constant.
> > Such an assumption holds if the induced distribution of the optimal policy $\pi^{*}$ is absolutely continuous with respect to $\nu$ and the space $\mathcal{S}\times\mathcal{A}$ is compact under the  topology induced by the Euclidean distance. Compared to the assumptions in [3,7,8], our assumption is **much weaker**, since the prior works [3,7,8] require that the ratio between the induced distribution of $\nu$ and those of all the other policies is upper bounded by a finite constant. We note that this assumption **cannot be further relaxed** in view of the impossibility result shown in Theorem 4 of [9]. Theorem 4 therein shows that the sample complexity of learning an $\varepsilon$- policy can not be polynomial in the size of the action space and the horizon.

---

> > > ### Author Response · Authors · 2022-08-02
> > > **Response to Reviewer gWc2: Part 3**
> > >
> > > * Assumption 3:  This assumption concerns the **realizability** of the transformer function class. Like Assumption 1, it holds when the function class has strong expressiveness. A similar assumption was made in [10]. It can be further relaxed to an **approximate realizability** assumption. However, making this weaker assumption induces an additional term in the suboptimality bound in Theorem 3, but it does not alter the bound’s dependence on the number of agents (which is one of our main contributions).
> > >
> > > * Assumption 4: Similar to Assumption 2, this assumption essentially characterizes the **distributional shift** between the visitation measure induced by the optimal policy and that induced by the behavior policy. More concretely, this assumption states that the total variation between the system dynamics and the functions in the function class computed along the trajectory induced by the optimal policy can be bounded by that induced by the sampling distribution $\nu$ up to a constant.
> > > It holds if the induced distribution of the optimal policy $\pi^{*}$ is absolutely continuous with respect to $\nu$ and the space $\mathcal{S}\times\mathcal{A}$ is compact. Compared to the corresponding assumptions in [11], it is **significantly weaker**, since [11] requires coverage for all the policies. Like Assumption 2, this assumption cannot be further relaxed for our theoretical results to hold. In particular, it is used in Step 3 in Line 737 in the proof of Theorem 3.
> > >
> > > We highlight that the present work **does not** seek to relax the assumptions in previous works; rather, our main contribution is the **theoretical demonstration** of the efficacy of the transformer in MARL, using previously agreed-upon assumptions in the literature [3,4,5].
> > >
> > > **Experimental results:** In the original submission, we presented the simulation results for the case where the number of agents $N$ is 15 in Appendix O. In Appendix O of our updated version of supplementary materials, we simulate the proposed algorithms on a cooperative navigation task in the Multiple Particle Environment (MPE) [12]. We present the table of the final rewards with different neural networks and different numbers of agents $N$ below. The **final rewards** reported are the averaged rewards reported over the last 10 training episodes. As shown in Theorem 1, the relational reasoning abilities of the deep sets and the Multi-Layer Perceptron (MLP) are worse than that of the set transformer. Hence, when the number of agents increases, the superiority of adopting the transformer to estimate the action-value function becomes more evident. This strongly corroborates our theoretical results in Theorems 1 and 2.
> > >
> > > |     &emsp; &ensp;	&nbsp; | &emsp; MLP &emsp;&nbsp;|Deep sets| Set transformer| &emsp; GCN  &emsp;&nbsp;|
> > > | &nbsp; -----&nbsp;	|&ensp;-----------&emsp;|&nbsp;-------------&thinsp;|&emsp;----------------	&thinsp;&emsp;|&emsp;----------&emsp;|
> > > | N=3  &nbsp;	| -218.66   &ensp;&ensp;| -219.97&thinsp; &ensp;| &emsp;-213.66 &ensp;&emsp;&emsp;| -234.56   &emsp;	|
> > > | N=6  &nbsp;	| -4096.53 &nbsp;	| -3476.76  	| &emsp;-2936.54 &emsp;&emsp;| -3731.03&ensp;&thinsp;	|
> > > | N=15&nbsp;| -3482.63  &ensp;| -1466.11  	| &emsp;-1390.90 &emsp;&emsp;| -1831.01&ensp;&thinsp;	|
> > > | N=30 	| -14751.74 	| -9112.87  	| &emsp;-7096.56 &emsp;&emsp;| -15728.25 	|
> > >
> > > **Connection with Online MARL**: We highlight that our main contribution concerns the **theoretical understanding** of the relational reasoning and the generalization abilities of the transformer. We demonstrate these for the offline MARL problem in this work. The generalization bound of the transformer in Proposition 2 can also be **plugged into the analysis of other types of RL settings**.
> > > For example, in the online RL problem with the bilinear model, Foster et al. [13] show that the regret can be upper bounded by the product of the Decision-Estimation Coefficient (DEC) and the model estimation error in Section 7.1.1 in [13]. If we adopt the transformer function class, this model estimation error **can then upper bounded with our results in Proposition 2**. For another example, in the online contextual RL problem, Zhang [14] shows that the regret can be upper bounded as the sum of the decoupling coefficient and the least square error in Equation (8) in [14]. Our generalization bound in Proposition 2 can also be conveniently **used to bound the least square error term**.
> > >
> > > [1] T. Zhou et al. Cooperative multi-agent transfer learning with level-adaptive credit assignment, arXiv preprint arXiv:2106.00517, 2021.
> > > [2] M. Wen, J. G. Kuba, R. Lin, W. Zhang, Y. Wen, J. Wang, and Y. Yang. Multi-Agent Reinforcement Learning is a Sequence Modeling Problem. arXiv preprint arXiv:2205.14953, 2022.
> > > [3] A. Antos, C. Szepesvári, and R. Munos. Learning near-optimal policies with Bellman-residual minimization based fitted policy iteration and a single sample path. Machine Learning, 71(1), 89-129, 2008.

---

> > > > ### Author Response · Authors · 2022-08-02
> > > > **Response to Reviewer gWc2: Part 4**
> > > >
> > > > [4] R. Munos, and C. Szepesvári. Finite-Time Bounds for Fitted Value Iteration. Journal of Machine Learning Research, 9(5), 2008.
> > > > [5] T. Xie, C. A. Cheng, N. Jiang, P. Mineiro, and A. Agarwal. Bellman-consistent pessimism for offline reinforcement learning. Advances in neural information processing systems, 34, 6683-6694, 2021.
> > > > [6] C. Yun, S. Bhojanapalli, A. S. Rawat, S. Reddi, and S. Kumar. Are transformers universal approximators of sequence-to-sequence functions? In International Conference on Learning Representations, 2020.
> > > > [7] M. Uehara, J. Huang, and N. Jiang. Minimax weight and q-function learning for off-policy evaluation. In International Conference on Machine Learning (pp. 9659-9668), PMLR, 2020.
> > > > [8] T. Xie, and N. Jiang. Batch value-function approximation with only realizability. In International Conference on Machine Learning (pp. 11404-11413), PMLR, 2021.
> > > > [9] J. Chen, and N. Jiang. Information-theoretic considerations in batch reinforcement learning. In International Conference on Machine Learning (pp. 1042-1051), PMLR, 2019.
> > > > [10] M. Uehara, and W. Sun. Pessimistic model-based offline reinforcement learning under partial coverage. In International Conference on Learning Representations, 2022.
> > > > [11] S. Ross, and J. A. Bagnell. Agnostic system identification for model-based reinforcement learning. In Proceedings of the 29th International Conference on Machine Learning, pages 1905–1912, 2012.
> > > > [12] R. Lowe, Y. I. Wu, A. Tamar, J. Harb, A. Pieter, and I. Mordatch, Multi-agent actor-critic for mixed cooperative-competitive environments. Advances in neural information processing systems, 30, 2017.
> > > > [13] D. J. Foster, S. M. Kakade, J. Qian, and A. Rakhlin. The statistical complexity of interactive decision making. arXiv preprint arXiv:2112.13487, 2021.
> > > > [14] T. Zhang. Feel-good Thompson sampling for contextual bandits and reinforcement learning. SIAM Journal on Mathematics of Data Science, 4(2), 834-857, 2022.

---

> > > > > ### Comment · Reviewer_gWc2 · 2022-08-08
> > > > > **Response to authors rebuttal**
> > > > >
> > > > > I would like to thank the authors for their thorough response to my questions/concerns. I found the explanation of the assumptions quite helpful and hope that a summary of that can be incorporated into a later version of the paper, making the presentation/significance clearer for a wider audience.
> > > > >
> > > > > Thanks for pointing me to the experimental results in the supplementary (unless I am mistaken these were not referenced from the main text?). While these results show that transformers indeed outperform other architectures the authors compared to, I am not sure if they strongly support the theoretical results of the paper, namely the generalisation error and sub-optimality gap being independent of the number of agents in the model free setting. The total reward achieved decreases with N for all models (which is maybe fine as that is not the subject of the bounds and the definition of the reward itself depends on N) but the experiment doesn't tell us much about how the value function approximation behaves with respect to N.

---

> > > > > > ### Author Response · Authors · 2022-08-09
> > > > > > **Re: Response to authors rebuttal**
> > > > > >
> > > > > > We thank the reviewer for raising these questions concerning experimental validations.
> > > > > > Let us reiterate the description and implications of this experiment here. We simulate the proposed algorithms on a cooperative navigation task in the Multiple Particle Environment (MPE), where $N$ agents move cooperatively to cover $L$ landmarks in the environment. Given the positions of $N$ agents $x_{i}\in\mathbb{R}^{2}$ for $i\in[N]$ and the positions of $L$ landmarks $y_{j}\in\mathbb{R}^{2}$ for $j\in[L]$, the agents receive the reward $r=-\sum_{j=1}^{L}\min_{i\in[N]}\parallel y_{j}-x_{i}\parallel_{2}$, which is almost **linear** in the number of landmarks $L$. Following the simulation setting in previous works, such as [1], we increase $L$ simultaneously with the number of agents $N$. This ensures that the total reward implicitly depends on $N$. After normalizing the reward by $L$, the average reward does not always decrease with $N$, as shown in the table below.
> > > > > > | &emsp; &emsp;&ensp;| &emsp;MLP &nbsp;| Deep sets 	| Set transformer 	| &ensp;GCN &ensp;&nbsp;|
> > > > > > |&ensp;------&ensp;|&ensp;---------&ensp;|&ensp;-----------&emsp;|&emsp;------------------&emsp;&thinsp;|&ensp;---------&ensp;|
> > > > > > | N=3  &ensp;| -72.89  &ensp;| &ensp;-73.32  &ensp;&nbsp;&thinsp;	| &emsp;&emsp;-71.22 &emsp;&emsp;&ensp;| &nbsp;-78.19 &nbsp;|
> > > > > > | N=6  &ensp;| -682.76 	| -579.46 &ensp;&ensp;| &emsp;&emsp;-489.42 &emsp;&emsp;| -621.84 |
> > > > > > | N=15 	| -232.18 	| -97.74  &emsp;&ensp;| &emsp;&emsp;-92.73 &emsp;&emsp;&ensp;| -122.07 &thinsp;|
> > > > > > | N=30 	| -491.72 	| -303.76 &ensp;&ensp;| &emsp;&emsp;-236.55 &emsp;&emsp;| -524.28 	|
> > > > > >
> > > > > > [1] I. J. Liu, R. A. Yeh, and A. G. Schwing. PIC: permutation invariant critic for multi-agent deep reinforcement learning. In Conference on Robot Learning (pp. 590-602). PMLR, 2020.

---

> ### Author Response · Authors · 2022-08-08
> **Response to Reviewer gWc2**
>
> Dear Reviewer gWc2,
>
> Since the author-reviewer discussion period has started for a few days, we will appreciate if you could check our response to your review comments soon. This way, if you have further questions and comments, we can still reply before the author-reviewer discussion period ends. If our response resolves your concerns, we kindly ask you to consider raising the rating of our work. Thank you very much for your time and efforts!

---

### Meta-Review · Area_Chair_L2Yz · 2022-08-29

**Recommendation:** Accept
**Confidence:** Less certain

**Metareview:**

The paper presents theoretical results justifying the use of transformers in cooperative multi-agent RL. The authors demonstrate that, with this choice of architecture, sub-optimality gaps grow independently of the number of agents. The theoretical contribution seems strong, with a more limited experimental evaluation. The paper is dense mathematically and was hard to assess. The theorems were nevertheless deemed strong enough to justify acceptance, however please do address comments from reviewer gWc2 in the final version.

**Award:**

No

---

### Decision · Program_Chairs · 2022-09-14

Accept